# Isolation, (bio)synthetic studies and evaluation of antimicrobial properties of drimenol-type sesquiterpenes of *Termitomyces* fungi

Nina B. Kreuzenbeck[1], Seema Dhiman[2], Dávid Roman [1], Immo Burkhardt[3], Benjamin H. Conlon[4], Janis Fricke[1], Huijuan Guo [1], Janis Blume[2], Helmar Görls[5], Michael Poulsen [4], Jeroen S. Dickschat [3], Tobias G. Köllner[6], Hans-Dieter Arndt[2] & Christine Beemelmanns [1,7,8]✉

Macrotermitinae termites have farmed fungi in the genus *Termitomyces* as a food source for millions of years. However, the biochemical mechanisms orchestrating this mutualistic relationship are largely unknown. To deduce fungal signals and ecological patterns that relate to the stability of this symbiosis, we explored the volatile organic compound (VOC) repertoire of *Termitomyces* from *Macrotermes natalensis* colonies. Results show that mushrooms emit a VOC pattern that differs from mycelium grown in fungal gardens and laboratory cultures. The abundance of sesquiterpenoids from mushrooms allowed targeted isolation of five drimane sesquiterpenes from plate cultivations. The total synthesis of one of these, drimenol, and related drimanes assisted in structural and comparative analysis of volatile organic compounds (VOCs) and antimicrobial activity testing. Enzyme candidates putatively involved in terpene biosynthesis were heterologously expressed and while these were not involved in the biosynthesis of the complete drimane skeleton, they catalyzed the formation of two structurally related monocyclic sesquiterpenes named nectrianolins.

[1] Chemical Biology of Microbe-Host Interactions, Leibniz Institute for Natural Product Research and Infection Biology – Hans Knöll-Institute (HKI), Beutenbergstraße 11a, 07745 Jena, Germany. [2] Institute for Organic and Macromolecular Chemistry, Friedrich-Schiller-University Jena, Humboldtstr. 10, 07743 Jena, Germany. [3] Kekulé-Institute of Organic Chemistry and Biochemistry, University of Bonn, Gerhard-Domagk-Straße 1, 53121 Bonn, Germany. [4] Section for Ecology and Evolution, Department of Biology, University of Copenhagen, Universitetsparken 15 2100, Copenhagen, Denmark. [5] Institute for Inorganic and Analytical Chemistry, Friedrich-Schiller University, Humboldtstrasse 8, 07743 Jena, Germany. [6] Department of Natural Product Biosynthesis, Max Planck Institute for Chemical Ecology, Hans-Knöll-Straße 8, 07745 Jena, Germany. [7] Helmholtz-Institut für Pharmazeutische Forschung Saarland (HIPS), Helmholtz Zentrum für Infektionsforschung (HZI), Campus E8.1, 66123 Saarbrücken, Germany. [8] Universität des Saarlandes, Campus E8, 66123 Saarbrücken, Germany. ✉email: Christine.Beemelmanns@helmholtz-hips.de

Among the different types of nutritional symbiosis, fungi-culture in insects represents one of the most complex symbiotic interactions[1,2]. Only a few insect lineages maintain and manure fungi as nutritional ectosymbionts, similar to human agriculture, with attine ants and *Macrotermitine* termites being prime examples[1]. In the case of fungus-growing termites, the basidiomycete genus *Termitomyces* is propagated by termite workers within biomass-containing cork-like structures ("fungus comb") as food fungus, where the fungus colonizes the provided predigested plant material via a dense hyphal network. Subsequently, the fungus forms visible protein, carbohydrate-rich and spore-containing fungal nodules that the termites ingest (Fig. 1a)[3,4]. While most *Termitomyces* nodules serve as nutrition for the termites and vector fungal spores (asexual reproduction), other types of nodules occasionally differentiate to become pointy primordia of fruiting bodies (Fig. 1b)[5]. Once pointy primordia mature (Fig. 1c), mushrooms spread sexual spores, which are likely picked up by foraging termites enabling the inoculation of newly founded termite colonies. However, it has been argued that the formation of fruiting bodies likely wastes resources that could otherwise have been allocated to growth of the fungus within the colony. This has led to the hypothesis that termites might actively suppress fruiting body formation of their fungal symbiont by consumption of mushrooms at a primordial stage, and in response the fungus likely evolved gut-resistant asexual spores encapsulated in nodules to ensure its propagation[5]. Thus, it has remained a conundrum in the termite–fungus symbiosis how the reproductive interests of host and symbiont are aligned[1].

Due to the long co-evolutionary history and the intertwined lifestyles of termites and their food fungus, it has been hypothesized that within the below-ground farming termite system diffusible volatile organic compounds (VOCs) of termites and fungus might play a key role as intra- and interspecific chemical mediators orchestrating the complex symbiosis[6–8].

While studies have showed profound influence of VOCs on behavior, development, and physiology of some social insect species, studies on fungus-growing insects, and particularly termites, have remained rather sparse. Similarly, our understanding of why and when fungi release specific VOCs[9–12], and their effects on fungal growth, sexual life cycle[13], and fungal interactions remains fragmented[14–16]. In the fungus-growing termite *Odontotermes obesus*, a first study indicated that termites are likely able to differentiate between the scent of the food fungus and ascomycetous fungal garden antagonist *Pseudoxylaria*, and that volatiles emitted by the antagonist, such as the sesquiterpenes aristolene and viridiflorol, are likely triggers of termite hygiene measures[17]. We recently commenced to analyze the VOC profiles emitted by laboratory cultures of *Termitomyces cryptogamus*[18] retrieved from healthy *Macrotermes natalensis* colonies[19]. To our surprise, only few terpenoid features were detectable in the volatilome of *Termitomyces* cultures maintained under laboratory growth conditions, although comparative genomic studies indicated that the fungal mutualist encodes above average numbers of e.g., terpene cyclases responsible for the biosynthesis of e.g., volatile terpenoids[20,21].

To test the hypothesis that the biosynthesis of terpenoid-based VOCs is life stage dependent and thus tightly regulated in *Termitomyces*, we analyzed the VOC pattern of *Termitomyces* in their natural environment and in different growth stages. This unveiled distinct VOC patterns across different biosample types. In mushrooms, sesquiterpenes appeared as dominant features, with drimenol (**1**) as one of the recurring features emitted from

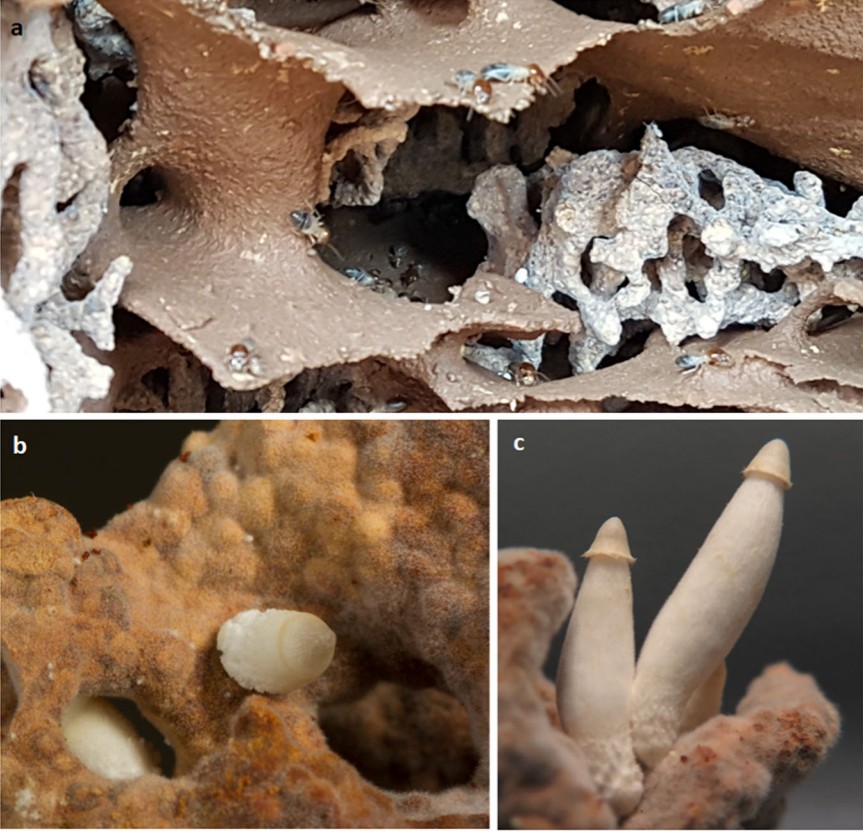

**Fig. 1 Fungus comb environment of a *Macrotermes natalensis* colony. a** Underground fungus garden chambers in which the fungus comb is maintained by grooming termite workers. **b** Differentiated pointy nodule of *Termitomyces* (*Termitomyces cryptogamus*). **c** Young mushroom developing from pointy nodules after 3 days.

fungal cultures. Targeted purification allowed the isolation of five drimenol derivatives and their characterization by NMR and x-ray crystallography analyses. Total synthesis of a focused drimane library was then pursued and allowed for a comprehensive evaluation of production levels and bioactivity patterns. Subsequent mining of genomic and transcriptomic data uncovered three candidate terpene synthase genes putatively involved in the biosynthesis of drimenol-like terpenes. Heterologous expression of enzyme candidates and bioassays yielded two monocyclic sesquiterpenes, putative biosynthetic shunt products of the drimenol or isolongifolene pathway. Data from our comprehensive study further define how fungal communication might occur by volatiles in the farming symbiosis with termites.

## Results and discussion

### *T. cryptogamus* mushrooms and fungus comb exhibit different VOC pattern.

We collected four different types of samples: (1) fungus-comb interspersed with *Termitomyces* mycelium, (2) fungus-comb from which a 4-day old *Termitomyces* mushroom emergesd (3) mushrooms separated from fungus-comb, and (4) axenic fungal agar plate cultures of *Termitomyces* sp. 153 (closest relative: *Termitomyces cryptogamus*) from which volatiles were captured on activated charcoal filters using the closed-loop-stripping analysis (CLSA) technique[22,23]. VOCs from mushrooms, fungus comb, and agar plate were collected using a CLSA apparatus over a period of 24 h (Figs. S2–S6, Tables S1-S4). The obtained headspace extracts were analyzed using gas chromatography-mass spectrometry (GC-MS), and data sets were dereplicated and putatively annotated using the National Institute of Standards Mass Spectral Library (NIST 2017). The detected VOCs emitted from comb, mushroom, and axenic cultures were distinct from each other (Fig. 2). Mushroom samples emitted only few aliphatic features with 3-octanone (communication signal, antifungal)[24], octan-3-ol (phytotoxic and antibacterial), oct-1-en-3-ol[12] (communication signal) and 2-nonenal (antifungal)[25] as the major detectable lipidic constituents. In contrast, a distinct terpenoid signature with sesquiterpenes as the most abundant compound class was detectable within the emitted blend of mushrooms. Overall, β-barbatene, β-cubebene, and brasiladienes were the most dominant features, while drimenol[26], intermedeol, african-1-ene, and α-amorphene were detected in lower abundances. Here, it was intriguing to note that prior studies on nodules reported the emission of a mostly monoterpene-dominated volatile blend (e.g., α-pinene[27], camphene, limonene).

In contrast to mushrooms, the volatile blends of fungus comb samples were dominated by aliphatic acids (e.g., hexanoic acid, isovaleric acid, pentanoic acid), ketones (e.g., 3-octanone, 2-undecanone), alcohols (e.g., hexanol), and furan derivatives (e.g., 2-pentylfuran, γ-lactones), while β-barbatene was the only detectable terpene feature. The VOC profile changed again when *T. cryptogamus* isolates were grown on potato dextrose agar (PDA) with drimenol and isolongifolene amongst the most abundant sesquiterpenes, in addition to 2,5-diisopropylpyrazine[28] and 1,2,4-trimethoxybenzene, the methylated derivative of the fungal redox-factor[20]. Intrigued by the emission of the antimicrobial and insect antifeedant sesquiterpene drimenol (1)[29], we monitored the composition of the volatile blend during fungal growth of *Termitomyces* sp. T153 on three different media (PDA, cellulose medium, fungus-comb medium) by GC-MS/MS (Figs. S7–S10, Tables S5–S7). Overall, emission levels of drimenol (1) appeared to reach a maximum after three to four weeks independent of the cultivation medium, and were still detectable even after seven weeks of growth. Interestingly, we were able to assign most of the detectable sesquiterpenes to putative *Termitomyces*-specific terpene cyclase (TTC) encoding genes (see Table S21)[21].

### *T. cryptogamus* isolates produce drimanes and drimenols.

In addition, secreted metabolites were extracted and analyzed by LC-MS/MS to detect less volatile sesquiterpenoid features during growth on solid media. These studies allowed the identification of more than five molecular ion features with typical sesquiterpenoid MS-fragmentation patterns, which were also detectable by stable isotope feeding experiments using $^{13}C$-enriched acetate (Fig. S11). To clarify their structural assignment, MS- and NMR-guided purification of extracts derived from *Termitomyces* sp. T153 (PDA for 14–21 days) was pursued, which yielded in total five drimenol derivatives (Fig. 3). Chemical structures of isolated compounds were elucidated mainly by comparative 1D and 2D NMR and MS/MS-analysis (Tables S8–S17). The $^1H$-NMR spectrum of compound 2 showed three main $CH_3$ singlets ($\delta_H$ 0.85, 0.86, and 0.98) and one singlet with slightly higher chemical shift ($\delta_H$ 1.77) what suggests its attachment to a double bond. This theory was underpinned by finding of only one olefinic methine at $\delta_H$ 5.53. The main octahydronaphtalene core of the molecule was built up based on the mutual $^1H$-$^1H$ COSY correlations between H-1 and H-2, H-2 and H-3, H-5 and H-6, H-6, and H-7. Additional finding of oxy-methine ($\delta_H$ 3.25) was located to C-3 based on $^1H$-$^1H$ COSY correlation with H-2. Position of hydroxymethyl group ($\delta_H$ 3.73 and 3.85) was solved from $^1H$-$^1H$ COSY correlation between H-9 and H-11 and from $^1H$-$^13C$ HMBC correlation of H-11 to C-8. Further $^1H$-$^13C$ HMBC analysis revealed a correlation of H-3 to C-14 and C-15, H-1 to C-13, and H-7 to C-12 what helped to assign an exact position of $CH_3$ groups and led to complete structure elucidation. X-ray crystallography of a single crystal of compound 2 ($CH_2Cl_2$/MeOH) secured the structural assignment (Fig. 3). Compounds 3 and 4 showed a similar chemical shift pattern. The main difference was observed in higher chemical shifts of protons at C-1 ($\delta_H$ 2.74 and 2.28) and C-2 ($\delta_H$ 2.22 or 2.29 and 1.56) which was caused by a presence of a neighboring keto group at C-3 concluded from appearance of a quaternary carbon with $\delta_C$ 216.7 for 3 and 215.8 for 4. Presence of a second hydroxymethyl group ($\delta_H$ 4.38 and 4.43) in 4 was deduced from $^1H$-$^13C$ HMBC correlations between H-12 and C-9, H-7 and C-12. Closer NMR analysis of compound 5 revealed similar structure core than 2 with hydrofuranone ring in position C-8 and C-9. For compound 6, a dehydroxylated core in C-3 with $\delta_H$ 1.22 and 1.41 was deduced. No presence of a double bond in $^1H$-NMR was observed. However, a chemical shift of H-7 ($\delta_H$ 3.76) and $\delta_C$ 72.9 for C-7 and $\delta_C$ 76.0 for C-8 suggested that the alkene underwent a dihydroxylation. Structural assignment of isolated compounds 2-6 was further supported by comparison of analytical data with literature reports (compound 2: *Marasmius oreades*[30], compound 3: *Clitocybe conglobate*[31], compound 4: *Phellinidium sulphurascens*[32], compound 5: *Peniophora polygonia*[33], compound 6: synthesis of stereoisomers[34,35] and synthetic derivatives (*vide infra*).

### Synthesis of drimane and drimenol derivatives.

To deduce the absolute stereochemistry of isolated compounds, to cover a broader range of a putative metabolome, and to establish bioactivity patterns (see below), we synthesized a focused library of drimanes and drimenol-like derivatives. We first explored a semi-synthetic approach (Fig. 4a). The synthesis commenced with the degradation of commercially available (+)-sclareolide (7) to the intermediate 8-hydroxy-drimenol (9) following a 3-step literature protocol[36] using Baeyer–Villiger oxidation as a key step. An acid-catalyzed regioselective dehydration of the tertiary alcohol provided drimenol (1), which was diastereoselectively hydrogenated in the presence of $PtO_2$ as the catalyst to C8-β-methyl drimanol (12) in 95% yield[37]. Diol 9 was subjected to $Ph_3P$–$I_2$ mediated dehydration yielding conjugated diene 10[38]. Finally, drimenal (11) was obtained from drimenol (1) by Swern oxidation (77% yield).

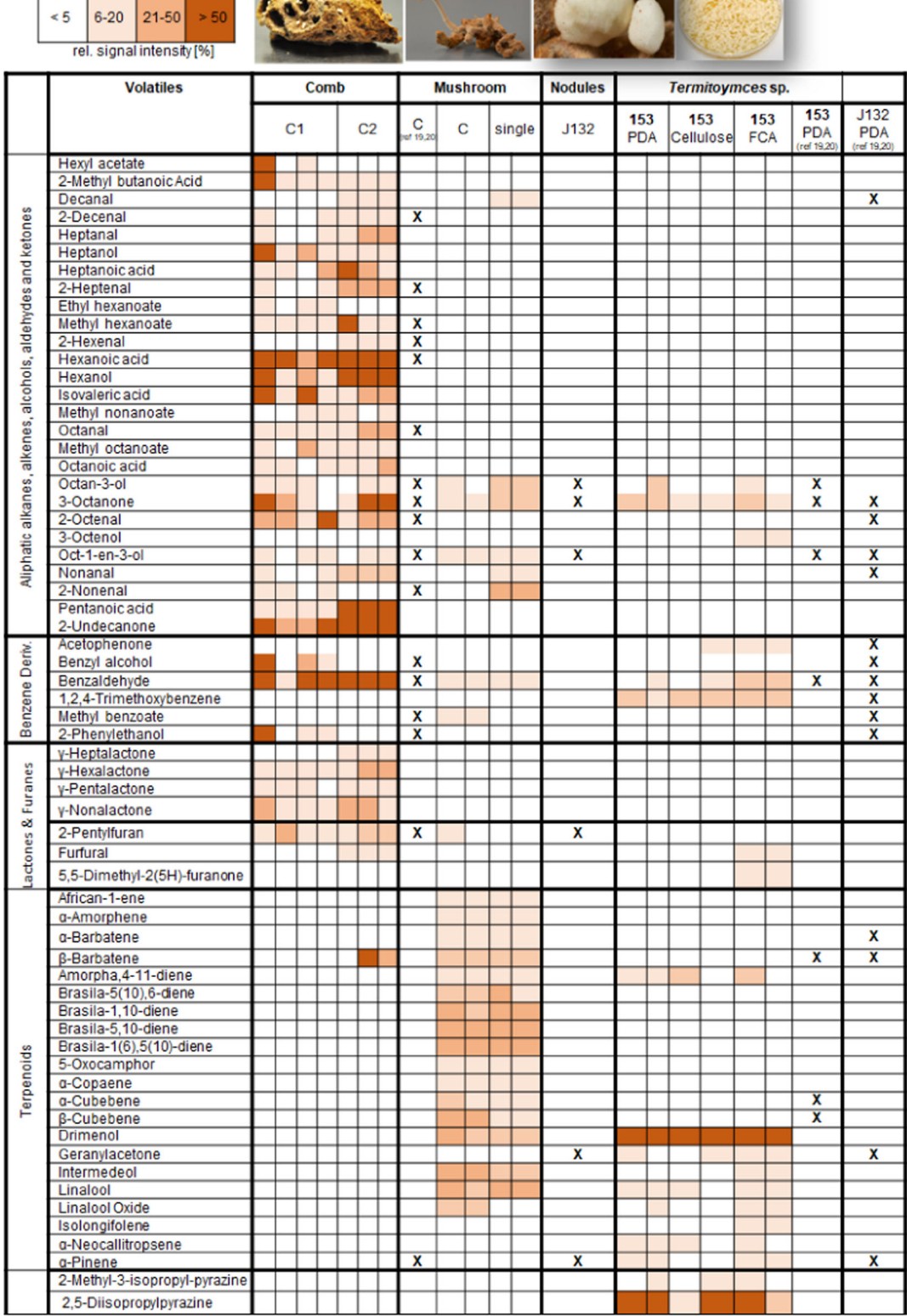

**Fig. 2 Heatmap of emitted volatiles from fungus comb (C), *T. cryptogamus* mushroom (M), and axenic lab cultures of *T. cryptogamus* strain T153.** Cultures of *T. cryptogamus* sp. T153 were grown on different solid media (PDA, Cellulose-Agar, FCA). Samples were measured as biological replicates with each column representing one replicate. Chemical features are depicted in relative signal intensities (based on the most abundant signal) using a color code. Reports of chemical feature in previous studies are marked with a bold **X**[20,21].

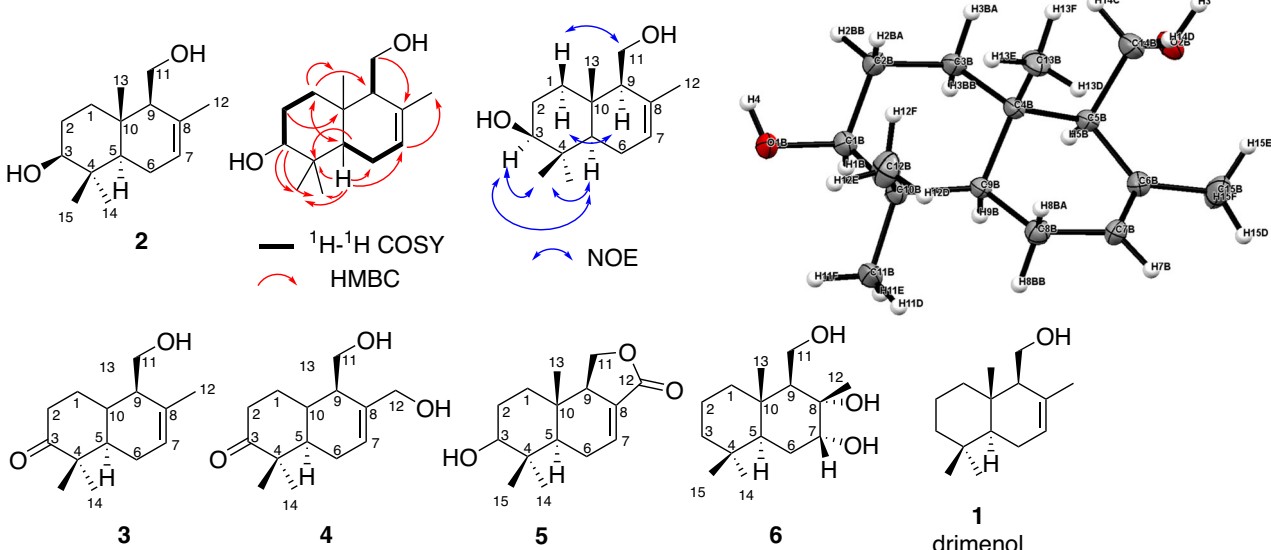

**Fig. 3 Chemical structures of detectable and isolated sesquiterpenes 1, 3-6 from *T. cryptogamus* strain T153 with atom numbering.** Drimenol derivative **2** with key COSY, HMBC and NOE correlations. Molecular structure with ellipsoids represents a probability of 30 %, H atoms are drawn with arbitrary radii, carbon atoms = gray, oxygen atoms = red, hydrogen atoms = white.

The C3-oxygenated drimanol derivatives **18** and **19** were then synthesized from (−)-(*S*)-10,11-epoxyfarnesyl acetate (**15**) by employing a Ti(III)-catalyzed cyclization cascade as a key step (Fig. 4b)[39–41]. Epoxyfarnesyl acetate (**15**) was obtained from *trans,trans*-farnesol (**13**) by using a literature procedure (Fig. S13)[42–44]. Acetylation of farnesol (**13**) followed by an asymmetric Sharpless dihydroxylation with AD-mix-β delivered diol **14** in 30% yield and 97% *ee*. The enantiomeric excess of **14** was determined by [1]H NMR analysis of its corresponding mono-(*S*)-MTPA ester. Selective mesylation of the secondary alcohol in diol **14** followed by ring closure with $K_2CO_3$-MeOH generated epoxide **15**, which was subjected to Ti(III)-catalyzed cyclization cascade conditions to obtain alkene **16**[39]. Subsequent hydrogenation using Pd/C as catalyst furnished the single diastereoisomer **17** in quantitative yield. Subsequent cleavage of the acetate group under basic conditions afforded 3-*S*-hydroxy-drimanol (**18**) in 90% yield, while Dess-Martin-Periodinane mediated oxidation of **17** followed by deprotection of acetate generated 3-oxo-drimanol (**19**). Comparative analysis of spectroscopic data of synthesized and isolated metabolites indicated similar chemical shifts patterns which correlated to the same bicyclic core structure (Tables S8–S19).

By using the isolated and synthetic compounds as metabolomic references, we then tested if drimenol derivatives were also produced by two other *Termitomyces* isolates. After two weeks of growth on PDA, we found the characteristic GC-MS and HRMS/MS signatures of compounds (**1-6**) in isolates T112 and J132 (Figs. S12, S13), but we were unable to identify any matching signal for derivatives **7-19** in organic extracts of *Termitomyces* cultures. Thus, we assume that only a subset of drimenols might be characteristic natural products for members of the fungal genus *Termitomyces*.

**Identification and analysis of putative drimenol synthases.** Drimane- and drimenol-type sesquiterpenes are a large group of natural products containing a $C_{15}$ bicyclic skeleton that have been identified from various plants[45], animals, the two major division within the fungal kingdom (Basidiomycota and Ascomycota)[26,46,47], and bacteria[48,49]. The biosynthesis of drimenol (**1**) has also gained much attention as drimenol can be formed via both class I and class II catalytic mechanisms. While drimenol synthases identified from plants were characterized as class I terpene cyclases (TCs), bacterial drimenol synthases were identified as class II sesquiterpene cyclases[50–52]. In contrast, dedicated bifunctional haloacid dehalogenase-like (HAD-like) terpene cyclases were reported from *Aspergillus* species[53,54] and most recently from *Antrodia cinnamomea*[55]. Hence, we performed a phylogenomic survey of members of the genus *Termitomyces* to identify putative candidate sequences, but were unable to relate any of the terpene synthase sequences identified in our previous study to reported sequences involved in the drimenol or isolongifolene biosynthesis (Tables S20, S21)[21]. We then performed a manual BLAST search against predicted protein sequences from different *Termitomyces* genomes using the sequence of the characterized HAD-like terpene cyclase AstC protein sequence (Gene ID AORIB40_05908) from *Aspergillus oryzae* as query sequence. In total, we identified three orthologous sequences (DS1-3), which contain motifs of type I and type II terpene synthases. While sequence DS1 is encoded in drimenol-producing *Termitomyces* strains T153, T112, and J132, sequences DS2 and DS3 were encoded in four (DS2) of a total of six *Termitomyces* genomes (Figs. S14–S18). We further validated our genome mining results by sequence alignments, which uncovered the characteristic conserved motifs (DDXXE, DxDTT, QW) important to perform ionization-dependent type I and protonation-dependent type II cyclisation reactions, thus indicating bifunctional properties of the enzyme (Figs. S14–S18). Notably DS3-T153 (57.2 kDa) lacked a hydrophobic C-terminal sequence, which was present in the sequence of DS1-T153 (59.5 kDa) and DS2-T153 (66.0 kDa).

We then revisited the putative formation of oxidized sesquiterpenes **2-6**[31,56], and hypothesized that either drimenol (**1**) or drimenyl pyrophosphate should serve as their biosynthetic precursor (Fig. 5)[57]. For compound **2**, an enzymatic hydroxylation of **1** at C-3, appeared most likely, which would yield after a second oxidation step, similar to what was recently shown in *Aspergillus calidoustus*[54] and *A. cinnamomea*[55]. At this stage, we also considered that epoxy-FPP **21** could serve as a substrate for

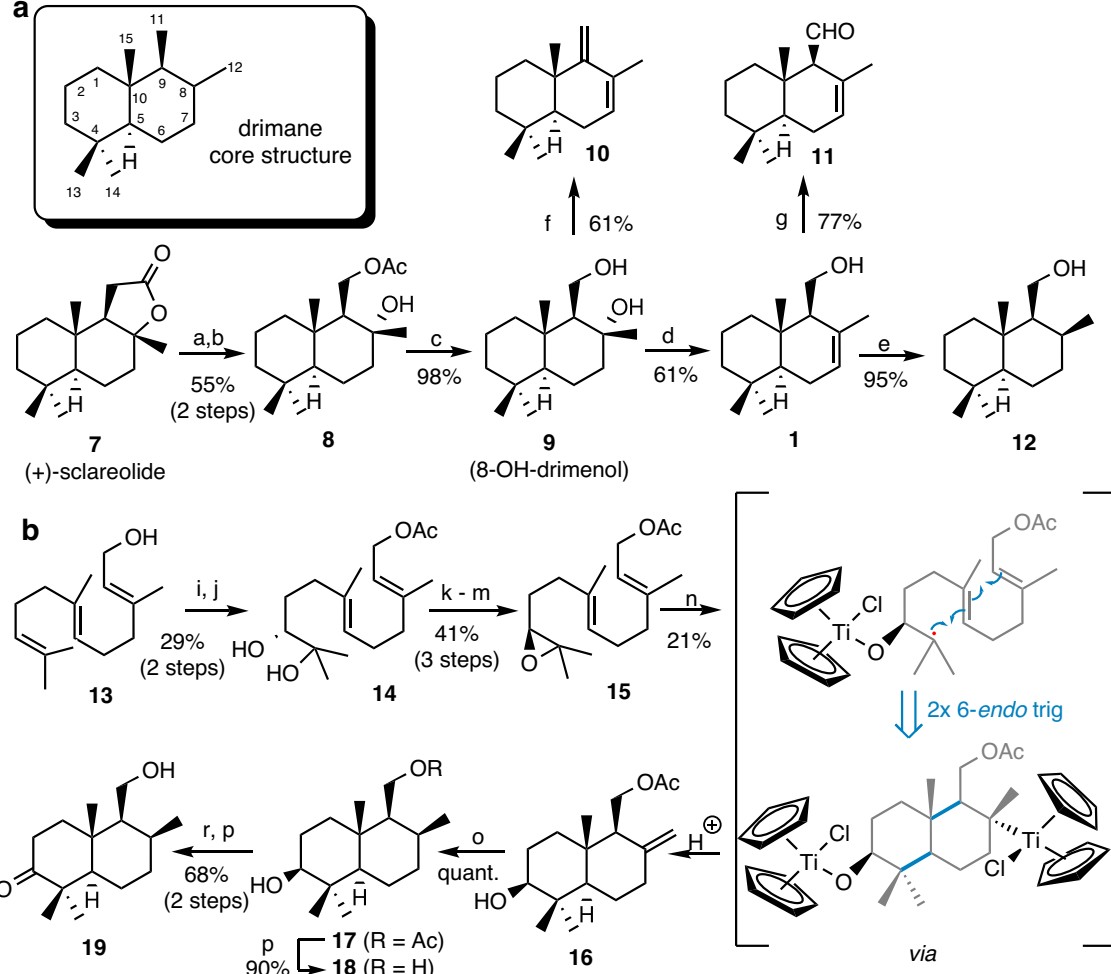

**Fig. 4 Total synthesis of drimenol, drimanol, and drimane scaffolds. a** Synthesis of drimenol (**1**), drimanol (**12**) and drimanes (**10** and **11**) from (+)-sclareolide (**7**). **b** Titanocene-catalyzed synthesis of 3-hydroxy-drimanol (**18**) and 3-oxo-drimanol (**19**) with highlighted key step intermediates. Reagents and conditions: a) MeLi, Et$_2$O, −78 °C, 30 min, 84%; b) (CF$_3$CO)$_2$O, H$_2$O$_2$, NaHCO$_3$, CH$_2$Cl$_2$, 0 °C, 2 h, 66%; c) KOH, MeOH, r.t., 10 min, 98%; d) TsOH*H$_2$O, CH$_2$Cl$_2$, 25 °C, 6 h, 61%; e) H$_2$, PtO$_2$, EtOAc, 25 °C, 2 h, 95%, dr 96:4; f) PPh$_3$, I$_2$, CH$_2$Cl$_2$, 25 °C, 4 h, 61%; g) (COCl)$_2$, DMSO, NEt$_3$, CH$_2$Cl$_2$, −78 °C to 25 °C, 4 h, 77%; h) (COCl)$_2$, DMSO, NEtiPr$_2$, CH$_2$Cl$_2$, -78 °C to 25 °C, 4 h, 40%; i) Ac$_2$O, DMAP, NEt$_3$, CH$_2$Cl$_2$, 0 °C, 30 min, 98%; j) AD-mix β, MeSO$_2$NH$_2$, tBuOH/H$_2$O (1:1), 24 h, 0 °C, 30%, 97% ee; k) MsCl, NEt$_3$, CH$_2$Cl$_2$, 0 °C, 20 min; l) K$_2$CO$_3$, MeOH, 25 °C, 30 min, 45% (2 steps); m) Ac$_2$O, DMAP, CH$_2$Cl$_2$, 25 °C, 1 h, 93%; n) Cp$_2$TiCl$_2$, Mn (powder), THF, 25 °C, 20 min, then 2,4,6-collidine, TMSCl, THF, **13**, 25 °C, 16 h, 21%; o) Pd/C, H$_2$, MeOH, 25 °C, 12 h, quant.; p) K$_2$CO$_3$, EtOH, 25 °C, 4 h, 90%; r) DMP, CH$_2$Cl$_2$, 25 °C, 3 h, 68% (2 steps).

**Fig. 5 Putative biosynthesis of drimenol in Termitomyces from FPP via drimenyl PP.** The reaction requires enzymatic protonation at C3 via a class II terpene synthase mechanism, followed by dephosphorylation of drimenyl PP by a class I terpene synthase reaction pathway.

the yet unknown DS, which would result—after cyclization and hydrolysis—in the formation of compound **2** as has been shown for the fungal meroterpenoid terretonin[58]. Subsequent enzymatic oxidation of diol **2** at C-12 could then yield drimentriol (**II**), which after additional oxidation at C-3 affords **4**. Similarly, a putative drimentriol **II** might also serve as precursor for aldehyde **III**, which could then undergo an oxidative transformation to

lactone **5** via hemiketal **IV** formation[59]. To gain insights into the observed enzymatic oxygenation patterns, we also surveyed the genomic environment up- and downstream (30 kbp) of the putative DS sequences in *Termitomyces* sp. T153. However, neither orthologous of previously described cytochrome P450 enzymes nor other oxidoreductases were detectable (Tables S22–S24).

**Fig. 6 Putative biosynthetic formation of drimane skeleton and drimenol congeners.** Detected derivatives **1-6** from cultures of *Termitomyces* spp. are highlighted in blue boxes. A putative drimenol synthase [DS] is proposed to catalyze the cyclization of farnesyl pyrophosphate (FPP) or epoxy-FPP to yield drimenol (**1**) or 3-hydroxy drimenol (**2**). Enzymatic oxidative transformations [O] and hydrolytic steps are proposed to catalyze the transformation to the identified oxidized drimane and drimenol derivatives via proposed intermediates (**I–V**).

Previous RNAseq data analysis of different substrates (fungus comb and *Termitomyces* mycelium from agar plates)[20,21] clearly indicated that gene sequences of DS1-3-T153 were actively transcribed and coinciding with drimenol production (Fig. S19, Table S25). Thus, the gene sequence of DS2 was obtained after amplification from the respective cDNA (*Termitomyces* sp. T153) sequence (Tables S25–S27), while codon-optimized transcripts (*Termitomyces* sp. T153) of DS1 and DS3 were synthesized (BioCat GmbH). Subsequent heterologous production of the histidine-fusion (His6) proteins was performed in *E. coli* BL21(DE3) using a pET28a vector. However, only DS3-His6 (57.2 kDa), which lacked a hydrophobic C-terminal sequence, was obtained as soluble protein. Removal of the hydrophobic tail in DS1 and DS2 afforded only insoluble protein fractions, but subsequent introduction of the maltose-binding protein (MBP) tag to the N-terminus of both sequences yielded finally soluble protein (DS1-MBP (102 kDa) and DS2-MBP (105.1 kDa; Figs. S20–S22)). Enzyme assays were performed using these purified proteins (DS1-MBP, DS2-MBP and DS3-His6) in the presence of MgCl$_2$ and with FPP (**20**) as substrate. While DS1-MBP and DS2-MBP showed no cyclization activity (Figs S23, S24), incubation of DS3-His6 with FPP (**20**) afforded detectable amounts of two sesquiterpene alcohols **22** and **23** (Fig. 6). However, neither retention time nor fragmentation pattern correlated with data obtained from isolated or synthesized derivatives or the NIST 2017 database. The enzyme reaction was also performed in different buffer systems (Tris-, phosphate-, citrate-buffer) at pH levels of 6 to 8 and in the presence of Mg$^{2+}$ or Co$^{2+}$ as cofactor, which resulted in all cases in similar product ratios, while only the addition of Mn$^{2+}$ as cofactor inhibited the enzyme reaction (Figs. S25–27). In a previous study on *Termitomyces* terpene cyclase 15 (TTC15-T153) from strain

T153, the characterized enzyme was versatile in substrate acceptance and product profile[21]. Thus, we included GPP and GGPP as well as FPP-epoxide and farnesol-epoxide as epoxidized substrates can be cyclized by type II terpene synthases (e.g., oxidosqualene cyclase) resulting in a hydroxy group at the respective position[60]. However, pyrophosphates GPP, GGPP, and FPP-epoxide yielded only dephosphorylated starting material, while farnesol-epoxide remained unreacted (Fig. S27).

To determine the chemical structures of the heterologous products **22** and **23**, the enzyme reaction was performed again with 58 mg purified DS3-His6 enzyme (enzyme derived from a 4 L induced *E. coli* BL21 (DE3) pET28a (+) culture, Fig. S28) in the presence of FPP, which yielded after 24 h and MS-guided purification the respective volatile compounds **22** (0.3 mg) and **23** (0.9 mg). The relative structures of **22** and **23** were deduced from 1D and 2D-NMR analysis. The $^1$H NMR spectrum of compound **22** showed four diagnostic methyl groups ($\delta_H$ 0.86, $\delta_H$ 0.87, $\delta_H$ 1.61 and $\delta_H$ 1.68), two olefinic methines at $\delta_H$ 5.41 and $\delta_H$ 5.42, and an oxy-methine ($\delta_H$ 4.15), which correlated to olefinic $^{13}$C signals at $\delta_C$ 122.4, 122.8, 139.7 and 141.1 (Tables S28–S29). The existence of a cyclohexene core was supported by detectable reciprocal $^1$H-$^1$H COSY correlations between H-7, H-8, H-9 and H-10. Compound **23** showed a very similar chemical shift pattern with the only difference in the chemical shift of one methyl group and a new characteristic CH signal at $\delta_H$ 1.42 (Tables S30–S31). Dereplication of chemical shift patterns and 2D NMR correlations indicated that compounds **22** and **23** exhibited the same planar monocyclofarnesyl skeleton as the natural product nectrianolin C. Based on the comparative analysis of their chemical shifts and optical rotations, the absolute stereochemistry was determined as (*6R,7S*)-**22**. For compound **23**, to the best of our knowledge, only the racemic product ***rac*-23** has so far been

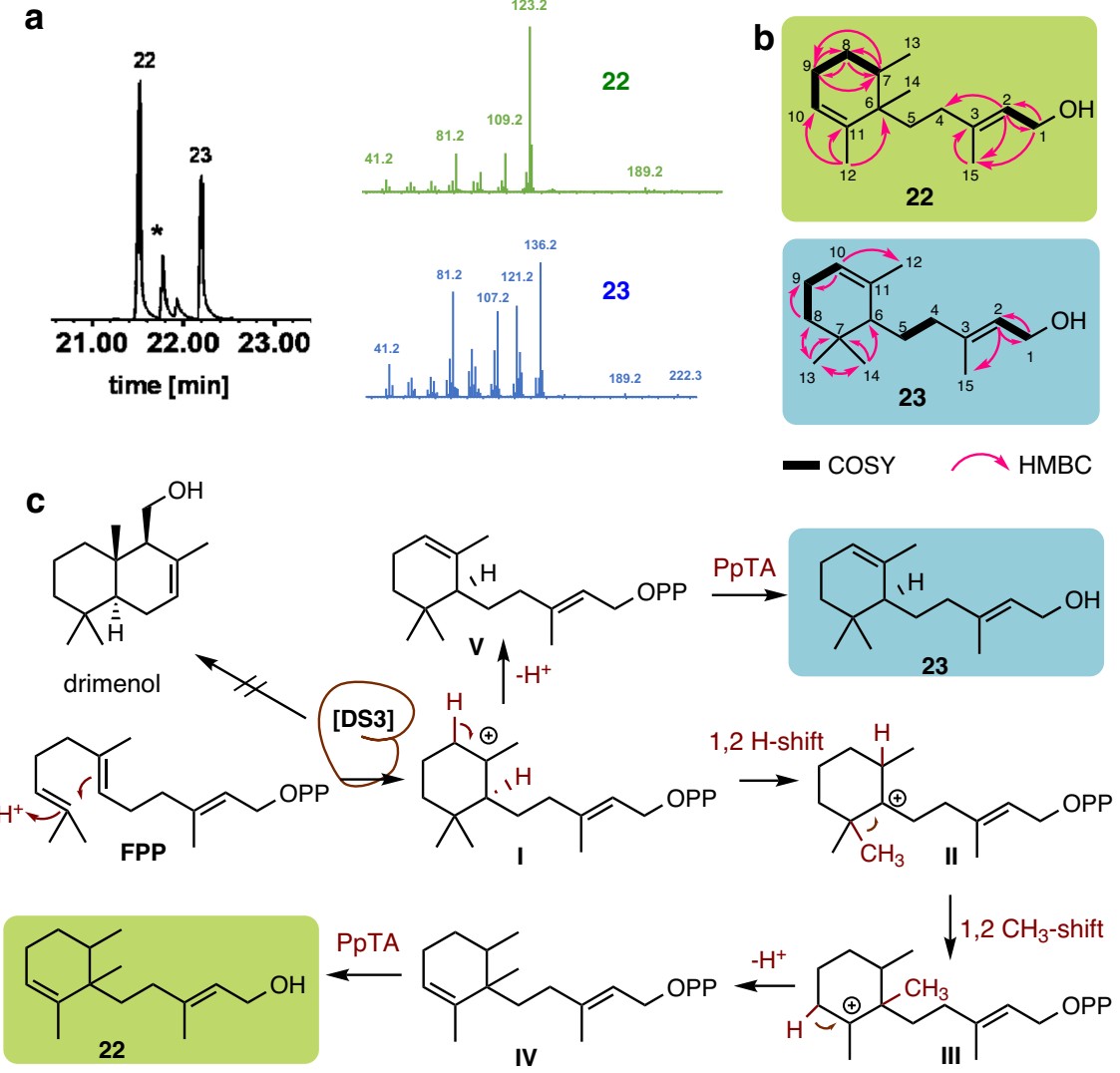

**Fig. 7 Enzyme assays with purified DS3-His6 protein yielded monocyclofarnesyl derivatives. a** GC-chromatogram of product spectrum after incubation of DS3 with FPP in the presence of $MgCl_2$ and $MS^2$-spectra of products **22**, **23**, and side product farnesol (*). **b** Key $^1$H-$^1$H COSY and HMBC correlations of isolated compounds **22** and **23**. **c** Proposed DS3-His6 catalyzed cyclisation of FPP and subsequent catalytic steps (in dark red) by enzyme DS3-His6 and a homolog of a phosphatidic acid phosphatase (PpTA), which result in the formation of products **22** and **23**.

reported[61], thus the absolute configuration of **23** (here named as nectrianolin D) remains elusive.

Nectrianolins were discovered in 2017 from the phytopathogenic fungus *Nectria pseudotrichia*[62], and show structural relation to (*R*)-*trans*-γ-monocyclofarnesol that was originally isolated from medicinal mushroom *Phellinus linteus*[55,63] and include several synthetic compounds[64,65]. As the candidate enzyme DS3-His6 offered neither drimenol nor any related $C_{15}$ bicyclic skeleton, we deduced that DS3-His6 likely acts either as a (*R*)-*trans*-γ-monocyclofarnesol synthase (similar to AncA) or as a dysfunctional drimenol synthase with similar mechanistic activity (Fig. 7). In addition, at this stage it cannot be fully excluded that phosphorylated precursor of **22**-**23** could serve as precursor for drimenols through a subsequent enzymatic reaction as originally proposed for (+)-*trans*-γ-monocyclofarnesol[63]. With isolated compounds at hand, we then reanalyzed if *Termitomyces* produced monocyclofarnesol volatiles in detectable amounts. However, we were unable to identify any matching signal for nectrianolins or (*R*)-*trans*-γ-monocyclofarnesol within the prior recorded MS-spectra, suggesting that these compounds are indeed likely to be transient biosynthesis intermediates.

**Drimenols show structure-dependent antibacterial activity**. Due to the widespread occurrence of drimane- and drimenol-type sesquiterpenes in nature, it is reasonable to assume that they fulfill important, yet mostly unknown, eco-physiological roles to the producer[46–49]. Furthermore, many derivatives display biologically and pharmacologically interesting activities such as anti-inflammatory, cytotoxic or antimicrobial activity[66,67], as well as antifeedant activities against insects in case of drimane dialdehydes[68]. Thus, we evaluated the activity of the compounds described in this study using ecology-based assays. However, we observed neither growth modulating effects on the producer *Termitomyces* sp. T153 (Fig. S29), nor was any antifeedant activity against the model organism *Spodoptera littoralis* observed[69]. We then evaluated their activity against a panel of fungal and bacterial test strains and were intrigued to note that drimenol **1** and aldehyde derivative **11** were most active, with moderate antifungal activities against *Penicillium notatum* and *Candida albicans* (Fig. 8), while antibacterial activity was only observed for derivatives **1** and **2** against *Staphylococcus aureus*, *Pseudomonas aeruginosa*, and *Mycobacterium vaccae*.

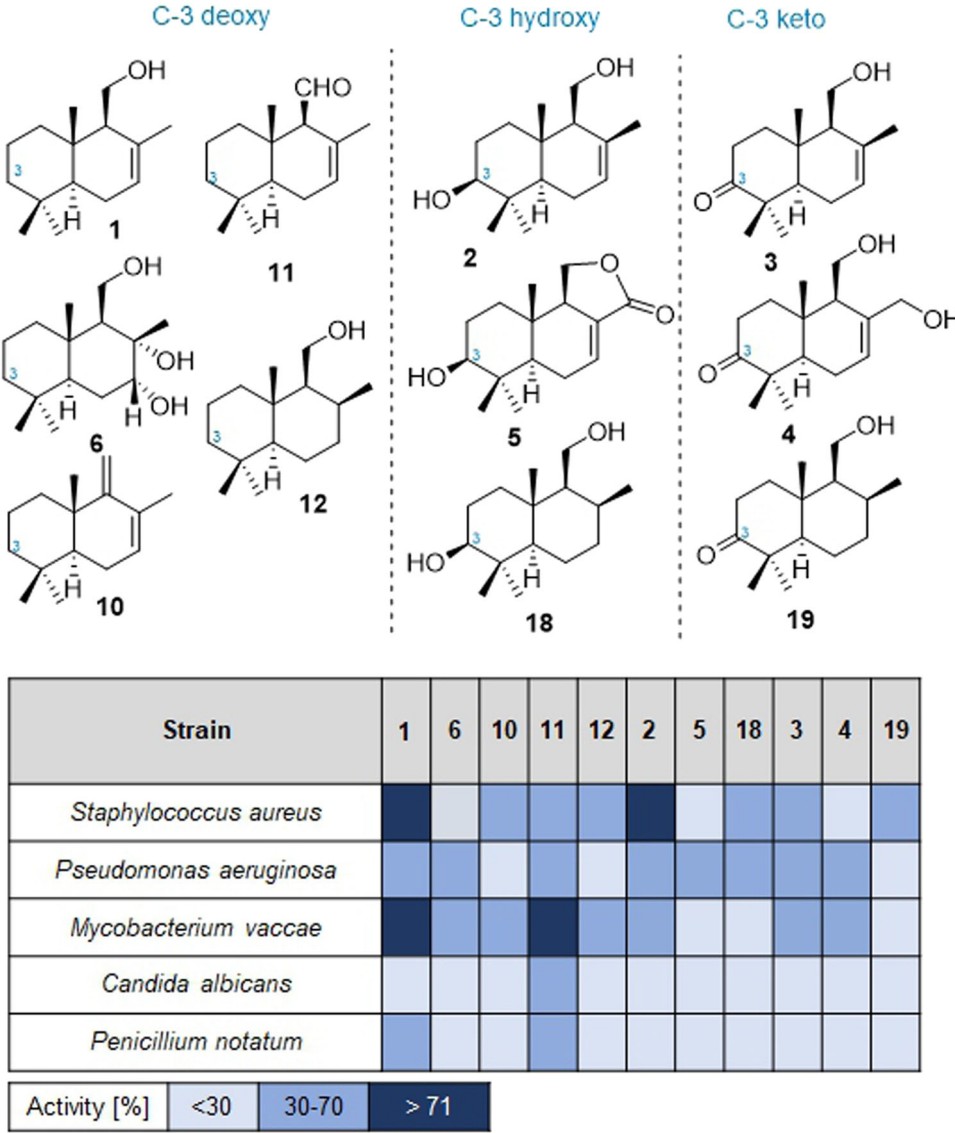

**Fig. 8 Antimicrobial activity test of drimane and drimenol derivatives.** Synthesized and isolated compounds (1 mg/mL dissolved in MeOH) were tested in agar diffusion assay against different bacteria and fungi. Positive controls: ciprofloxacin 5 μg/mL in dest. $H_2O$, amphotericin 10 μg/mL in DMSO/MeOH, Activity (ZOI [mm]) is reported as relative values compared to positive controls.

## Conclusion

Driven by our goal to entwine the volatile-based communication activities within the complex multi-partner symbiosis of the fungus-growing termite *M. natalensis*, we comparatively analyzed the emitted volatiles blends of mushrooms of different *Termitomyces* isolates associated with this termite species, along with fungus comb and axenic cultures. We demonstrated that mushrooms produce characteristic VOC patterns, which differ from nodules and axenic lab culture of the fungal mutualist. The combination of targeted isolation and synthesis of ten drimenol derivatives allowed us to verify the detected sesquiterpenoid features and study their antimicrobial effects, with drimenol (**1**) and isolated derivative (**2**). RNAseq-assisted analysis led to the identification of three putative drimenol synthase candidates. However, only one active enzyme variant (DS3-His6) was obtained by heterologous enzyme production, and which was found to catalyze the formation of cyclofarnesyl derivatives nectrianolin **22** and **23**.

Drimane sesquiterpenes exhibit diverse biological activities (e.g., cytotoxic, antifeedant, insecticidal, antimicrobial) and this

broad activity spectrum could play a key role in the fungus-growing termite system by mediating communication between termites and their fungal crop, but also by regulating growth of other microbes than *Termitomyces* sp. in the fungus comb. This is likely to be context-dependent; for example, drimenol was mainly emitted from mushrooms and lab cultures. This metabolite exhibits antifungal and germination inhibiting activity[29], which could aid in nest defense under natural conditions. As mushroom formation in nature has not yet been observed for fungal strains actively farmed by *M. natalensis* colonies, we hypothesize that termite workers may be able to detect morphology-dependent changes in the volatilome of their mutualistic fungus, including drimenols, and might respond by actively suppressing the energy-consuming formation of mushrooms by grooming or feeding activities. However, once *Termitomyces* mushrooms manage to develop in the absence of termites, the emission of this anti-microbial volatile blend may aid in protection from bacterial infection of the fruiting body. The biochemical findings of this study provided additional evidence for the importance of terpenes in this symbiotic relation and pave the way to unravel the

biochemical basis and function of sesquiterpene formation in *Termitomyces*.

## Material and methods

**Collections**. Fungus comb material was excavated from mature *Macrotermes natalensis* colony MN187 (S24 40.434 E28 48.275) and fungus comb carrying pointy nodules were collected from *Macrotermes natalensis* colony MN188 (S24 40.512 E28 48.260). Fungus combs were carefully transferred to sterile plastic containers supplied with a wet filter paper, where they were kept at 4 °C in the dark. Containers were regularly inspected for contaminations with non-*Termitomyces* fungi[5,20,21]. *Termitomyces* mushrooms grew from pointy nodules.

**Volatile analysis**. The emitted metabolites of all samples were collected shortly after collection using the Closed Loop Stripping Apparatus (CLSA) headspace technique (Fig. S2)[70,71]. Samples were placed in a closed chamber under circulating air stream passed through a charcoal filter (Chromtech GmbH, Idstein, Precision Charcoal Filter, 5 mg) for 24 h. The filter was extracted using 50 µl $CH_2Cl_2$ and the extracts were analyzed by GC-MS. Identification of compounds was performed by comparing the mass spectra to NIST spectra libraries and retention indices ($I$) from the literature. Structures were only annotated if both identifiers fit (Figs. S1–S9, Tables S1–S9).

**Cultivation**. *Termitomyces* spp. were cultivated on potato-dextrose agar (PDA, 25 mL per plate, standard 15 × 90 mm) for a maximum of four weeks at room temperature. Sub-culturing was done by scraping mycelium from half a plate, mixing with 10 mL sterile PBS and spreading 500 µL suspension per plate or used as inoculum of liquid culture broth. Three minimal media containing different carbon sources were inoculated with mycelium and cultivated at room temperature for a given time.

**Cultivation on $^{13}C$-isotope enriched medium**. *Termitomyces* sp. T153 was grown on reduced PDA medium (8.7 g/L) supplemented with 20 mM $^{13}C$-enriched sodium acetate (1-$^{13}C$ or 1,2-$^{13}C$, Sigma-Aldrich, USA) for three weeks. Afterward plates were cut into small pieces and extracted with $CH_2Cl_2$ overnight. The solvent was filtered and dried under vacuum. Crude extracts were dissolved in MeCN and analyzed using HR-LC-MS (Figure S10).

**Isolation procedures**. *Termitomyces* sp. T153 was cultivated on 40 PDA agar plates (2 L PDA) at room temperature for two and four weeks, respectively. After 2 weeks of incubation, 20 plates were cut in small pieces and soaked in $CH_2Cl_2$ overnight. Extracts were filtered and organic solvent evaporated. The crude extracts were dissolved in cyclohexane and fractionated on Chromabond $SiO_2$ column with a gradient from 100% cyclohexane to 100% EtOAc for 10 min using flash chromatography. A second purification step was performed using reverse phase HPLC with a phenyl-hexyl column and a gradient starting at 40% MeCN isocratic for 5 min followed by a gradient from 40% MeCN to 50% MeCN for 14 min. Final purification of compound **4** was performed using a phenyl-hexyl column and an isocratic gradient with 35% MeCN/65% $H_2O$ + 0.1% FA for 20 min. Compounds **5** and **6** were purified using a Luna C18 column and 35% MeCN/ 65% $H_2O$ + 0.1% FA or 45% MeCN/65% $H_2O$ + 0.1% FA, respectively.

After 4 weeks, the remaining plates were cut in small pieces and soaked in MeOH overnight. After filtration and solvent evaporation, organic extracts were redissolved in 20% MeOH and prepurified by solid phase extraction (SPE; 10 g, Macherey Nagel,

Germany) starting from 20% MeOH and eluting by increasing MeOH concentration in 10% steps to 100% MeOH. The 60% MeOH fraction was further purified by reverse phase HPLC on a Luna C18 column starting at 62.5% MeOH for 5 min, applying a gradient from 62.5% MeOH to 100% MeOH for 18 min. Pure compound **3** and compound **2** were obtained after additional using a Luna5u Phenyl-Hexyl column (Phenomenex, 250 × 10 mm) and an isocratic gradient of 45% MeCN over 23 min.

Compound **2**: colorless solid; $[\alpha]_D^{21.8}$ −4.78 (c 0.25 w/v%, $CHCl_3$); UV (MeCN/$H_2O$/FA) $\lambda_{max}$ 220, 230 nm; IR (ATR) $\nu_{max}$ 3744, 3381, 2925, 2855, 1700, 1456, 1386, 1165, 1022; 912, 842; HR-MS (ESI) $m/z$ [M-$H_2O$ + H]$^+$ Calcd for $C_{15}H_{25}O$ 221.18999 (Figs. S30, S31); Found 221.18951, NMR data: see Tables S8, S9 and Supplementary Data 1 (Figure NMR-S3 to NMR-S10).

Compound **3**: light-yellow solid; $[\alpha]_D^{21.6}$ −54.2° (c 0.37 w/v%, $CHCl_3$); UV (MeCN/$H_2O$/FA) $\lambda_{max}$; IR (ATR) $\nu_{max}$ 3444, 2968, 2856, 1698, 1450, 1385, 1037, 1011; HR-MS (ESI) $m/z$ [M + H]$^+$ Calcd for $C_{15}H_{25}O_2$ 237.18491; Found 237.18449 (Figs. S32, S33), NMR data: see Tables S10, S11 and Supplementary Data 1 (Figure NMR-S12 to NMR-S18).

Compound **4**: light yellow solid; $[\alpha]_D^{22.0}$ -23.51° (c 0.23 w/v%, $CHCl_3$); UV (MeCN/$H_2O$/FA) $\lambda_{max}$; IR (ATR) $\nu_{max}$ 3362, 2923, 2858, 1701, 1456, 1385, 993, 754; HR-MS (ESI) $m/z$ [M + H]$^+$ Calcd for $C_{15}H_{25}O_3$ 253.17949; Found 253.17982 (Figs. S34, S35); NMR data: see Tables S12, S13 and Supplementary Data 1 (Figure NMR-S19 to NMR-S28).

Compound **5**: light-yellow solid; $[\alpha]_D^{21.6°}$ −6.63 (c 0.06 w/v%, $CHCl_3$); UV (MeCN/$H_2O$/FA) $\lambda_{max}$ 230 nm; IR (ATR) $\nu_{max}$ ($CHCl_3$) 3854, 2923, 2853, 1749, 1457, 1388.5; HR-MS (ESI) $m/z$ [M + H]$^+$ Calcd for $C_{15}H_{23}O_3$ 251.16417; Found 251.16383 (Figs. S36, S37), NMR data: see Tables S14, S15 and Supplementary Data 1 (Figure NMR-S29 to NMR-S37).

Compound **6**: colorless solid; $[\alpha]_D^{22.0}$ −8.0 (c 0.17 w/v%, $CHCl_3$); UV (MeCN/$H_2O$/FA) $\lambda_{max}$; IR (ATR) $\nu_{max}$ 3750, 2922, 2868, 1362, 1074, 756, 665; HR-MS (ESI) $m/z$ [M + H]$^+$ Calcd for $C_{15}H_{29}O_3$ 257.21112; Found 257.21051 (Figs. S38, S39), NMR data: see Tables S16, S17 and Supplementary Data 1 (Figure NMR-S38 to NMR-S49).

**X-ray analysis**. The intensity data were collected on a Nonius KappaCCD diffractometer, using graphite-monochromated Mo-$K_\alpha$ radiation. Data were corrected for Lorentz and polarization effects; absorption was considered on a semiempirical basis using multiple scans[72–74]. The structure was solved by direct methods (SHELXS) and refined by full-matrix least-squares techniques against Fo$^2$ (SHELXL-2018). The hydrogen atoms bonded to the hydroxyl groups O1, and O2 of the two independent molecules A and B of **2** were located by difference Fourier synthesis and refined isotropically. All other hydrogen atoms were included at calculated positions with fixed thermal parameters. All non-hydrogen atoms were refined anisotropically[75]. MERCURY was used for structure representations[76].

Crystal Data for **2** (Supplementary Data 2): $C_{15}H_{26}O_2$, $M = 238.36$ g mol$^{-1}$, colorless prism, size 0.112 × 0.110 × 0.098 mm$^3$, orthorhombic crystal system, space group P ī, $a = 9.98070(10)$, $b = 12.4732(2)$, $c = 21.7124(3)$ Å, $V = 2703.00(6)$ Å$^3$, T = -140 °C, Z = 8, $\rho_{calcd.} = 1.171$ gcm$^{-3}$, µ (Mo$K_\alpha$) = 0.71073 Å, trans. min.: 0.7057, trans. max.: 0.7456, $F(000) = 1056$, 21390 reflections in h(−12/11), k(−16/16), l(−28/28), measured in the range 2.25° ≤ Θ ≤ 27.48°, completeness $\Theta_{max} = 99.9\%$, 6186 independent reflections, $R_{int} = 0.0537$, 5641 reflections with $F_0 > 4\sigma(F_0)$, 331 parameters, 0 restraints, $R1_{obs} = 0.0472$, $wR2_{obs} = 0.1200$, $R1_{all} = 0.1031$, $wR2_{all} = 0.2385$, GOOF = 0.985, Flack-parameter −0.3(5), largest difference peak and hole 0.251/−0.186 e Å$^{-3}$.

**Organic synthetic approaches**. For experimental details, see Supplementary Methods within the Supplementary Information (Supplementary Note 1 (Synthesis of drimenol derivatives) and Supplementary Note 2 (Synthesis of FPP and derivatives). For NMR spectra of synthetic compounds, see Supplementary Data 1.

**Genome mining**. To improve the genome assembly and annotation of *Termitomyces* sp. T153 (GCA_018296165.1) the strain T153 was sequenced using Oxford Nanopore Technology (Oxford Nanopore Technologies, Oxford, UK). For this, DNA was extracted from *Termitomyces* sp. T153 cultures grown in PDB for 1 week (28 °C, 300 rpm). The mycelium was filtered, frozen at -80 °C and lyophylized for 24 h. The freeze-dried material was ground to a fine powder and 10 mg was used for subsequent DNA extraction using the DNA plant kit (Qiagen). For better purity, DNA was precipitated with ice cold *i*PrOH, centrifuged and the pellet was washed with 70% EtOH and dried by compressed air. For analysis DNA was dissolved in water. The MinION sequencing library was prepared using the Rapid DNA sequencing kit (SQK-RAD4) according to the manufacturer. DNA sequencing was performed on a MinION Mk1B sequencing device equipped with a R9.4.1 flow cell, which was prepared and run according to the manufacturer. Nanopore sequencing raw data was generated using MinKNOW software version 4.0.20 (Oxford Nanopore Technologies) and was base-called and trimmed using Guppy version 4.2.2 (Oxford Nanopore Technologies). The resulting fastq files were filtered using Nanofilt. A hybrid de novo genome assembly, combining BGISeq and Oxford Nanopore data, was performed using MaSuRCA version 3.4.1. The resulting draft assembly was then polished with the accurate Illumina reads using the POLCA genome polisher. Putative drimenol synthases in predicted proteins of *Termitomyces* sp. T153 were identified from BLAST search against AstC protein (Gene ID AORIB40_05908). The genomic environment of putative drimenol synthase genes (DS1-3) in *Termitomyces* sp. T153 was analyzed for the presence of putative natural product biosynthetic enzyme genes 30.000 bp up- and downstream of the DS1-3 sequences (Tables S20–25).

**Generation of DS2 sequence**. We reanalyzed the relative expression levels of TC-related gene sequences in RNAseq data obtained from axenic *Termitomyces* sp. 153 and J132, as well as fresh and old fungus comb and nodules on which young workers feed[21]. The transcript sequence of DS2 was obtained by PCR from a cDNA template. RNA of *Termitomyces* sp. T153 was extracted from frozen mycelium of a culture grown on PDA plate cultures for ~2 weeks using the "Isolate II RNA plant" Kit (Bioline). cDNA was obtained using the following protocol: 2 µg RNA, 2 µL Oligo d(T)$_{23}$VN primer (NEB), 1 µL dNTP mix (25 mM each, biotech rabbit) and adjustment with water to a volume of 18 µL; incubation for 5 min at 65 °C, followed by addition of 6 µL RT buffer, 1 µL RT enzyme (Thermo, Maxima H Minus), 1 µL RNase-Out (Invitrogen) and 4 µL water; cDNA synthesis was performed for 2–3 h at 46 °C finalized by an inactivation step at 85 °C for 5 min.

**Cloning protocol**. Native cDNA of DS2 was used as template for PCR amplification. The restriction site *NheI/BamHI* was added to forward and the restriction site *Hind*III added to reverse primer sequence for ligation into pET28a (+) vector (Table S25). PCR amplification protocol: 5 µL HF buffer, 0.5 µL dNTP mix (NEB), 7.25 µL H$_2$O, 0.25 µL Phusion, 10 µL Primermix (FW and REV, 2.5 µM each), 2 µL cDNA template with following reaction conditions: 95 °C 30 s, 35 cycles (98 °C 30 s, 61.5 °C 60 s, 72 °C 60 s, 98 °C 60 s), 72 °C 5 min. PCR templates were amplified at 61 °C.

DNA sequences were purified from agarose gel according to manufacturer's protocol of ZymoClean Gel DNA Recovery Kit. After PCR amplification and gel purification, PCR products were cloned in pJET vector according to manufacturer's blunt end protocol of CloneJET PCR Cloning Kit (Thermo). After transformation in *E. coli* DH5α cells, single colonies were screened for successful transformation by colony PCR. Plasmids were isolated from overnight cultures of transformed DH5α cells, with Innuprep Plasmid Kit (JenaAnalytik). Restriction digest of respective plasmids (1 µg) was performed with *NheI/BamHI* and *Hind*III (NEB) according to manufacturer's protocol. Fragments were ligated with T4 DNA ligase (NEB) starting from 22 °C to 16 °C (1 °C/1 h) and incubation for 10 h. For transformation, 2 µL of ligation mixture was used as described above. Codon-optimized DS1 and DS3 sequences in pET28a(+) vectors were ordered from Biozyme (Heidelberg). Vectors were transformed in chemically competent *E. coli* cells (DH5α for plasmid amplification, BL21 strain for heterologous protein expression) by heat shock (45 s) method. Plasmids were sequences with pET or pJET-specific primers at Eurofins Genomics.

**Cloning protocol for DS1-O and DS2-O**. Sequences of DS1-O and DS2-O were amplified for cloning in *pOPIN M vector* (Tables S26, S27) with following PCR reaction condition 95 °C/ 30 s, 35 cycles (98 °C/30 s, 61.5 °C/60 s, 72 °C/60 s, 98 °C/60 s), 72 °C 5 min. The vector was digested with *Kpn*I and *Hind*III (NEB) according to protocol and purified from agarose gel. Gibson Assembly (NEBuilder Hifi DNA assembly, NEB) was performed with vector-insert ratio 1:2 according to protocol and incubated for 1 h at 50 °C. The mixture was transformed in *DH5α*/BL21 cells and positive transformants were used for heterologous expression studies with ampicillin (100 µg/mL) as selecting antibiotic.

**Heterologous production of DS1-3**. An overnight culture of *E. coli* BL21 (DE3) containing the prepared plasmids was grown in LB medium (kanamycin 60 µg/mL). For expression experiments, a 1 mL o/n culture was used to inoculate 50 mL LB medium (0.4% glycerol, 60 µg/mL kanamycin). The culture was incubated first at 37 °C (180 rpm shaking) until a density of OD$_{600}$ 0.4, then incubated at 16 °C (160 rpm) for 1 h, and after that induced with 0.1 mM IPTG followed by growth over night (16 °C). Cells were centrifuged at 4 °C for 15 min (16.000 rpm) and resuspended in lysis buffer (2 mL, 200 mM NaCl, 100 mM Tris-HCl, 5% glycerol, pH 7.0). Homogenized cells were sonicated 3 × 2 min at 0 °C and again centrifuged as mentioned above. The supernatant was defined as *soluble fraction*. The pellet was resuspended in 1 mL urea lysis buffer (500 mM NaCl, 100 mM Tris-HCl, 7 M urea, pH 8) and incubated at room temperature for 5 min. After another centrifugation step, supernatant was transferred to a new tube and defined as pellet fraction. Samples were adjusted to a concentration of 5 mg/mL, diluted with 4× Laemmli sample buffer (Biorad), and incubated at 90 °C for 10 min. For SDS-PAGE, precast gels "Any kD Mini-PROTEAN TGX" (Biorad) were used.

**Protein purification**. Soluble protein fractions of heterologously expressed proteins were loaded on a Ni$^{2+}$-NTA-affinity column (Ni-NTA Agarose, Jena Bioscience) equilibrated with lysis buffer and binding was performed for 30 min at 4 °C with mild shaking. Afterwards the recombinant protein was washed on the column with washing buffer (2 × 20 mL, 500 mM NaCl, 100 mM Tris-HCl, 50 mM imidazole, 5% glycerol, pH 8) and eluted with 2.5 mL elution buffer (500 mM NaCl, 100 mM Tris-HCl, 300 mM imidazole, 5% glycerol, pH 8). The buffer of elution fraction was

exchanged to standard protein buffer (200 mM NaCl, 100 mM Tris-HCl, 5% glycerol, pH 7) by PD-10 desalting column (GE Healthcare).

**Western blot analysis**. an SDS-PAGE was transferred to a nitrocellulose membrane using the iBlot2 dry blotting system (ThermoFisher Scientific). Afterward, the membrane was shaken in blocking solution (TBS-T + 5% milk powder) for 1 h. Then, 40 μL of His-Probe antibody (HIS.H8, Santa Cruz Biotechnology) was added and incubated overnight under mild shaking. The resulting membrane was washed 3x with TBS-T, added to blocking solution and 10 μl of the secondary antibody (m-IgGκ BP-HRP, Santa Cruz Biotechnology). After 2 h shaking incubation, the membrane was washed and the chemiluminescence signal of HRP was visualized by adding WesternSure chemiluminescent substrate (LI-Cor Bioscience, according to protocol) and visualization with FUSION FX imaging system (Vilber Lourmat GmbH).

**GC-MS analysis of enzyme assay products**. Analysis of DS assay products was conducted using an Agilent 6890 Series gas chromatograph coupled to an Agilent 5973 quadrupole mass selective detector (interface temp, 270 °C; quadrupole temp, 150 °C; source temp, 230 °C; electron energy, 70 eV). Terpenes were separated using a ZB5 column (Phenomenex, Aschaffenburg, Germany, $30\,m \times 0.25\,mm \times 0.25\,\mu m$) and He as carrier gas (flow, 2 mL/min). The sample (1 μL) was injected without split at an initial oven temperature of 45 °C. The temperature was held for 2 min and then increased to 280 °C with a gradient of 7 °C/min, and then further increased to 330 °C with a gradient of 60 °C/min and a hold of 1 min.

**Enzyme assays**. Soluble protein fractions of DS1-MPB, DS2-MPB, His6-DS3, and a control culture carrying an empty pET28a(+) vector were prepared from 100 mL of *E. coli* and enzyme reactions were set up in glass vials. Terpene assays were performed using single purified enzymes (DS1-/DS2-MBP and His6-DS3) and combinations thereof. Enzyme assay using versions of DS1-DS3 was set up by mixing 50 μL soluble protein extract, 5 μL 10 mM FPP, 5 μL 200 mM MgCl$_2$, 40 μL assay buffer (10 mM Tris-HCl, 2 mM DTT, 10% glycerol, pH 7.2). Reaction mixture was covered with a layer of 100 μL hexane and incubated at 30 °C for 2 h. Samples were afterwards vortexed for 2 min, frozen in liquid nitrogen and hexane phase was taken directly for GC-MS analysis.

**Testing reaction conditions**. The assay was modified as follows: *pH optimum*: buffer of eluted protein fractions was exchanged to phosphate (50 mM, pH 6 and 7), citrate (50 mM, pH 5 and 6), or standard protein buffer (pH 7 and 8). *Cofactors*: The assay was modified as follows: 50 μL soluble protein extract, 5 μL 10 mM FPP, 5 μL 200 mM MgCl$_2$/0.5 mM MnCl$_2$/1 mM CoCl$_2$, 40 μL assay buffer (10 mM Tris-HCl, 2 mM DTT, 10% glycerol, pH 7.2) were mixed, covered with a layer of 100 μL hexane and incubated at 30 °C for 2 h (Figs. S25, S26). *Substrates*: Either 10 μL of 1 mg/mL GPP or GGPP, or 5 μL of 10 mM FPP/FPP-epoxide/ epoxyfarnesol were used (Fig. S27).

**Upscaling of enzyme assay**. Overnight culture of *E. coli* BL21 (DE3) pET28-DS3 (4 × 20 mL) were grown in LB (kanamycin 60 μg/mL) and used to inoculate 4 × 1 L LB medium (0.4% glycerol, 60 μg/mL kanamycin). The cultures were first incubated at 37 °C and 180 rpm shaking until the cells grew to a density of OD$_{600}$ 0.4, then kept at 16 °C (160 rpm) and induced with 0.1 mM IPTG after 1 h. After cultivation o/n

cells were centrifuged (4 °C, 15 min, 16.000 rpm) and resuspended in IMAC FPLC lysis buffer (15 mL per 1 L culture, 500 mM NaCl, 50 mM Tris-HCl, 50 mM imidazol pH 7.4). Homogenized cells were sonicated 3× for 2 min (0 °C) and centrifuged (4000 rcf, 30 min). The supernatant was purified by fast protein liquid chromatography (FPLC) on a NGC Quest 10 Plus Chromatography System from Bio-Rad with Nuvia IMAC Ni-Charged 5 mL column (BioRad) equilibrated with IMAC FPLC buffer. A gradient of 0% to 50% buffer B (50 mM Tris pH 7.4, 500 mM NaCl, 500 mM imidazole) was applied over 10 column volumes. Putative target fractions were checked by SDS-Page for correct size and potential impurities. Fractions containing protein DS3 were combined and concentrated with MWCO 5000 at 4000 rcf, 4 °C ca 3 h to a volume of 2 mL and diluted with standard protein buffer to 60 mL. Purified protein solution (60 mL) was diluted with assay buffer (48 mL) and incubated in the presence of FPP (115 mg synthesized FPP dissolved in 6 mL 50 mM NH$_4$HCO$_3$ buffer), and 200 mM MgCl$_2$ solution (6 mL), were incubated at 30 °C overnight. The reaction was quenched by addition of cyclohexane (120 mL), and then filtered through a short pad of Celite. The organic layer was separated and the remaining aqueous layer was extracted 3× with cyclohexane. Combined organic layers were washed with brine, dried over MgSO$_4$, filtered, and concentrated in vacuo to give 90.4 mg of the crude extract. Compounds were purified by flash chromatography (cyclohexane/EtOAc) and preparative HPLC purification (MeCN/ ddH$_2$O + 0.1% FA) and yielded compound **22** (0.3 mg) and compound **23** (0.9 mg) (Tables S28-S31 and Supplementary Data 1 (Figure NMR-S73 to NMR-S80)).

**Antimicrobial assay**. Disc diffusion assays of standard microbial strains were performed following the Clinical and Laboratory Standards Institute guidelines. Antimicrobial activities (zones of inhibitions) were monitored in mm and compared to positive and negative controls. For *Termitomyces* agar diffusion assays, PDA plates were inoculated with strain T153. After 1 day, paper discs soaked with 10 μl compound solution (1 mg/mL in MeOH) were placed in the middle of inoculated *Termitomyces* plates and incubated at room temperature. Plates and zones of inhibition were monitored daily for 12 days.

**Reporting summary**. Further information on research design is available in the Nature Portfolio Reporting Summary linked to this article.

## Data availability

All data generated or analyzed during this study are included in this published article and its supplementary files. Supplementary Information contains supplementary figures to experimental work, supplementary methods and notes. Supplementary Data 1 contains copies of all NMR spectra. Supplementary Data 2: Crystallographic Information File (CIF) for structure of compound 2 reported in this study. The X-ray crystallographic coordinates for structure of compound 2 reported in this study have also been deposited at the Cambridge Crystallographic Data Centre (CCDC), under deposition numbers CCDC-2175583. These data can be obtained free of charge from The Cambridge Crystallographic Data Centre via www.ccdc.cam.ac.uk/data_request/cif.

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

## Acknowledgements
We are thankful for support by Forestry and Agricultural Biotechnology Institute (FABI) and University of Pretoria (UP) to host research visits and provide logistic and laboratory infrastructure. We acknowledge funding by the Deutsche Forschungsgemeinschaft (DFG, German Research Foundation) under Project-ID 239748522—SFB 1127 to C.B. (A06) and H.D. (C03), and the possibility to acquire a 500 MHz NMR via an equipment grant (INST 275/442-1 FUGG). We also would like to acknowledge funding by the European Research Council (ERC-CoG - 771349) and The Danish Council for Independent Research (DFF - 7014-00178) to M.P. S.D. was funded by a Marie-Curie fellowship grant (THALLMORPHAL, no. 796194). Furthermore, we would like to thank Sabine Vreeburg for pictures of *Termitomyces*, Philipp Stephan (HKI) and Hajo Kries (HKI) for experimental assistance related to enzyme purification and provision of the pOPIN M vector, and Heike Heinecke (HKI) for NMR measurement.

## Author contributions
N.B., S.D., D.R., I.B., J.F., Hu.G.: designed research, N.B., S.D., D.R., I.B., J.F., Hu.G., J.B., T.K.: performed the experiments. N.B., S.D., D.R., I.B., B.H.C., J.F., J.B., T.K., Hu.G., C.B., H.D.A., J.D., M.P.: analyzed data. He.G. performed X-ray crystallography. N.B., S.D., D.R., B.H.C., J.F., Hu.G., He.G., C.B.: generated the figures. C.B., H.D.A., J.D., M.P.: wrote the manuscript with input from all authors.

## Funding

## Competing interests
The authors declare no competing interests. H.G. is an Associate Editor for *Communications Chemistry*, but was not involved in the editorial review of, or the decision to publish this article.
