## [Peer Review File · Communications Chemistry]

Reviewers' comments:

Reviewer #1 (Remarks to the Author):

The manuscript by Beemelmans and coworkers provides exceptional first insights into the VOC profiles of diverse *Termitomyces* vs. fungal comb samples. The work nicely combines analytical and synthetic organic chemistry, chemical synthesis, biochemical approaches and in vitro enzymatic characterization for the quite detailed characterization of drimenol-type sesquiterpenes (among many other VOCs detected). Overall, the paper pulls together a huge amount of very well designed and conducted experimental work to provide exciting insights into (changes of) the fungal VOC compositions, also characterizing individual antimicrobial properties of the target compounds. The study is a pleasure to read and - in my view - can be accepted virtually as is. Just two very minor suggestions:

- product yields are in part hard to find (for isolated and enzymatic products);
- some of the NMR spectra provided in the ESI would benefit from better phasing and/or baseline corrections (e.g., S110, S114, S116, etc.)

Reviewer #2 (Remarks to the Author):

Kreuzenbeck et al. studied the volatile organic compound (VOC) patterns emitted by fungus-growing termites from the Macrotermite family in a symbiotic relationship with the fungus *Termitomyces* and identified the key VOCs emitted by these symbionts in comb, mushroom, and axenic lab cultures. Interestingly, the mushroom samples and the axenic plate cultures of *Termitomyces* sp. 153 emitted significant amounts of sesquiterpenes, with drimenol, an antimicrobial and antifeedant compound, being the most abundant and recurring. The authors were also able to detect and identify less volatile sesquiterpenoid metabolites via LC-MS/MS and structurally characterised (via NMR, MS/MS, and X-ray crystallography) them to be five known drimenol derivatives. Through a library of synthesised drimanes, they were able to identify only a subset of drimenols to be characteristic natural products emitted by *Termitomyces* isolates. The authors, via genome mining, identified three putative drimenol synthases and carried out enzymatic assays with heterologously expressed and purified synthases to determine the products involved in the biosynthetic formation of the drimane skeleton. Only one of the synthases resulted in the production of two sesquiterpene alcohols, both of which, following structural characterisation, were determined to be nectrianolins rather than drimenols. Antimicrobial assays revealed moderate antibacterial activity for some of the drimenol derivatives.

While the authors present technically sound data to back their claims, particularly in terms of structural characterisation, they have not provided a convincing explanation of their findings in manuscript as it stands. However, the manuscript can be vastly improved by including relevant and key information pertaining to the background and previously known mechanisms associated with the study to help improve the understanding of a wider audience. A collated and detailed explanation of the results in a 'Discussion' section can help the reader navigate through the vast amounts of data presented. The manuscript is recommended for publication, following the revisions listed below.

1. A more detailed background is required explaining the specific symbiotic interactions between the termites and the fungi, including the roles played by VOCs. Additionally, information related to the importance of sesquiterpenoid compounds in this communication is essential for the reader to fully appreciate the relevance of the findings of the paper.

2. A schematic of the closed-loop stripping technique in the supporting information section would be useful for a broad audience.

3. Figure 2 explanation: What do M1, M2, and M represent? Additionally, the title for the figure specifies the colours brown, yellow, and blue which are absent in the heat map.
4. Previously reported information such as the occurrence, biological activities of a few isolated drimanes, and importance of the study of drimenol and drimane sesquiterpenoids with respect to their structures and how they connect lower and higher terpenoids, their use as potential biomarkers is required either in the Discussion section or while first introducing drimenol in the manuscript.
5. Figure 3 B), missing word: Chemical structures of isolated sesquiterpenes 1, 3-6.
6. How was the focused library of drimanes chosen for synthesis? Was it based on previous studies with the chosen drimanes suspected to be part of the metabolome?
7. Include a figure in the supporting information for the known biosynthetic mechanistic scheme of drimane sesquiterpenoids since the mechanism is different from majority of sesquiterpenoids. Scheme 2 begins with the formation of drimenol.
8. Explain the differences between ascomyceteous and basidiomycete fungi in the introduction paragraphs.
9. Typo in first line of the 'Identification and analysis of putative drimenol synthases' section associated with citation no. 47.
10. Include structure of isolongifolene and its relation to drimanes since several comparisons and references are made to the compound.
11. The following sentence should be changed along the lines of: 'In total, we identified three orthologous sequences (DS1-3) with sequence DS1 encoded in drimenol-producing *Termitomyces* strains T112 and J132, while sequences DS2 and DS3 were only encoded in four, in a total of six *Termitomyces* genomes studied (Figure S13-S16).'
12. Figure S19: Define what the lane 'C' represent on the SDS-PAGE gels? Also, mention the expected sizes of the proteins in the title or on the gels.
13. Page 9, 1st line: Reference to figure S21 is incorrect. S21 is the SDS-PAGE of FPLC fractions. It should only be S22.
14. Specify BL21(DE3) in the following sentence: 'enzyme derived from a 4 L induced *E. coli* BL21(DE3) pET28a (+) culture'.
15. Figure S25: Peaks are hard to distinguish. It looks like peaks corresponding to compounds 1 and 2 are seen in the negative control as well.
16. Include more information about nectrianolins in the Discussion section – relevance and relation to the field of study, it is important to understand their putative roles in sesquiterpene formation in *Termitomyces*.
17. Typo on page 9: '..it is also reasonable to speculate that 22-23 could share a phosphorylated precursor that might serve as intermediate in the biosynthesis of drimenol and isolongifolene (Figure 4C)', not Scheme 2.

18. Were additional substrate feeding experiments or gene deletion experiments done to conclude that compounds 22 and 23 are shunt products of the pathway?

19. A conclusion paragraph describing how drimenol and the studied drimanes (along with nectrianolins) can improve our understanding on the symbiotic relationship is recommended.

Reviewer #3 (Remarks to the Author):

Kreuzenbeck et al. describe the isolation, structure elucidation, chemical synthesis, and biological evaluation of the drimenol-type sesquiterpenes from *Termitomyces* fungi. Throughout the manuscript, it was clear and understood that the experimental methods and results strongly support the conclusions. If the authors revise the manuscript of their paper in consideration of the following points, I will consider it suitable for publication.

1. I think the title is somewhat inappropriate. I think that this title does not do a good job of understanding what is meant by "characterisation" and "antimicrobial properties". However, it is not a very serious issue. Please consider it.

2. In Introduction. The importance of VOCs is somewhat explained, but we thought that if the authors could actually mention examples of past "a key communication role", the point of focusing on VOCs in this study would be better conveyed.

3. In P3L64-74. Only spectral data are enumerated in SI, and the entire actual structure determination process is omitted. It seems to me that the structure determination process should be explained, if only a little.

4. In P5 Figure 3. If hydrogen is written in the structural formula, an arrow from hydrogen to HMBC correlation should be shown.

5. In P7 Scheme 2. I don't understand why there are two pathways coming up with FPP and epoxy-FPP as substrates. For each step of the conversion, they can only rely on the cited literature, and it is not clear how this is supported by past findings.

6. In P7 the explanation of Figure 4 (To test ... were detectable.). The description of the methods and results of bioinformatics analysis is too brief to understand the content. Please describe in more detail and carefully.

7. In P8. The description of the RNA-seq results is too brief to understand.

Point-to-point-response: **Isolation, (bio)synthetic studies and evaluation of antimicrobial properties of drimenol-type sesquiterpenes of *Termitomyces* fungi**

Reviewer #1 (Remarks to the Author):	
The manuscript by Beemelmans and coworkers provides exceptional first insights into the VOC profiles of diverse Termitomyces vs. fungal comb samples. [...]The study is a pleasure to read and - in my view - can be accepted virtually as is. Just two very minor suggestions:	
- product yields are in part hard to find (for isolated and enzymatic products);	We have included yields for isolated and enzymatic products within the manuscript (e.g. yield compound 22: 0.3 mg; compound 23: 0.9 mg)
- some of the NMR spectra provided in the ESI would benefit from better phasing and/or baseline corrections (e.g., S110, S114, S116, etc.)	All NMR spectra have been corrected and are now presented according to Commun. Chem. submission guidelines: one spectrum per page in a landscape format and tuned in order to achieve the best possible baseline correction.
Reviewer #2 (Remarks to the Author):	
While the authors present technically sound data to back their claims, particularly in terms of structural characterization, they have not provided a convincing explanation of their findings in manuscript as it stands. However, the manuscript can be vastly improved by including relevant and key information pertaining to the background and previously known mechanisms associated with the study to help improve the understanding of a wider audience.	We appreciate the very constructive comments and have revised part of the introduction to explain more in detail why volatile features are important factors when analyzing the intertwined life cycle of termites and their symbiotic fungus.
Introduction	
A more detailed background is required explaining the specific symbiotic interactions between the termites and the fungi, including the roles played by VOCs.	We have included additional sections within the introduction and discussion clearly explain the background and reasoning.
Additionally, information related to the importance of sesquiterpenoid compounds in this communication is	We have also added examples of the importance of sesquiterpenoids.

essential for the reader to fully appreciate the relevance of the findings of the paper.	
8. Explain the differences between ascomyceteous and basidiomycete fungi in the introduction paragraphs.	We have included more information of ascomyceteous and basidiomycete fungi in the introduction. We would like to point out that a stronger differentiation in this context might be rather confusing as both produce drimane-like terpenoids.
3. Figure 2 explanation: What do M1, M2, and M represent? Additionally, the title for the figure specifies the colors brown, yellow, and blue which are absent in the heat map.	M1 and M2 were replaced by C1 and C1, C stands for comb; colors were removed from subscription
4. Previously reported information such as the occurrence, biological activities of a few isolated drimanes, and importance of the study of drimenol and drimane sesquiterpenoids with respect to their structures and how they connect lower and higher terpenoids, their use as potential biomarkers is required either in the Discussion section or while first introducing drimenol in the manuscript.	We have included additional information on the occurrence of drimane and drimenol-type metabolites. Here we would like to add that more than hundred drimane-type sesquiterpenoids have been isolated. Hence, it is apparent that drimane derivatives are widespread in nature and deliver important eco-physiological roles to the synthesizers. However, it should be noted that drimane is rarely isolated as a natural product and that the simplest and closest drimane-type sesquiterpene found in nature is drimenol, which implies that drimenol or perhaps drimenyl pyrophosphate is the biosynthetic precursor to the various drimane sesquiterpenoids
5. Figure 3 B), missing word: Chemical structures of isolated sesquiterpenes 1, 3-6.	Corrected
6. How was the focused library of drimanes chosen for synthesis? Was it based on previous studies with the chosen drimanes suspected to be part of the metabolome?	As stated in the manuscript, "To deduce the absolute stereochemistry of isolated compounds, cover a broader range of a putative metabolome, and for establishing bioactivity patterns". We agree that from a synthetic point of few several more derivatives might have been interesting to synthesize. However, to balance synthetic efforts and targeted metabolomics efforts, we selected the depicted examples, which cover drimane as well as drimenol scaffolds.

9. Typo in first line of the 'Identification and analysis of putative drimenol synthases' section associated with citation no. 47.	corrected
10. Include structure of isolongifolene and its relation to drimanes since several comparisons and references are made to the compound.	We have included the structure into Scheme 2.
11. The following sentence should be changed along the lines of: 'In total, we identified three orthologous sequences (DS1-3) with sequence DS1 encoded in drimenol-producing Termitomyces strains T112 and J132, while sequences DS2 and DS3 were only encoded in four, in a total of six Termitomyces genomes studied (Figure S13-S16).'	Thank you, we have changed this accordingly.
16. Include more information about nectrianolins in the Discussion section – relevance and relation to the field of study, it is important to understand their putative roles in sesquiterpene formation in Termitomyces.	We have included a section to their putative origin and how this could relate to monocyclofarnesol and drimenols.
17. Typo on page 9: '..it is also reasonable to speculate that 22-23 could share a phosphorylated precursor that might serve as intermediate in the biosynthesis of drimenol and isolongifolene (Figure 4C)', not Scheme 2.	corrected
18. Were additional substrate feeding experiments or gene deletion experiments done to conclude that compounds 22 and 23 are shunt products of the pathway?	We very much appreciate the question. We have not yet performed additional experiments on the biosynthesis of 22 and 23, and conclusions are based on enzyme similarity to reported fungal drimenol synthases. However, we agree that subsequent study, following on this report is necessary to fully deduce drimenol and the biosynthesis of 22/23. Yet, we consider these detailed biochemical and molecular engineering-focus study currently beyond the scope of this manuscript.
19. A conclusion paragraph describing how drimenol and the studied drimanes (along with nectrianolins) can improve our	Thank you for the comment. We have included a statement to clarify the connections. As nectrianolins were not annotated features of the VOCs datasets measured from environmental

understanding on the symbiotic relationship is recommended.	or fungal samples, we concluded that either DS3-His6 is a dysfunctional drimenol- synthase producing the products by chance, and/or has similar activity as a monocyclofarnesol synthase or the yet uncharacterized nectrianolin synthase.
Supporting Information	
7. Include a figure in the supporting information for the known biosynthetic mechanistic scheme of drimane sesquiterpenoids since the mechanism is different from majority of sesquiterpenoids. Scheme 2 begins with the formation of drimenol.	Thank you for the suggestion, we have included a respective figure showing the following steps: biosynthetic pathway: 1) Protonation at C3 2) Double bond rearrangement and deprotonation 3) Cleavage of diphosphate group
12. Figure S19: Define what the lane 'C' represent on the SDS-PAGE gels? Also, mention the expected sizes of the proteins in the title or on the gels.	We have defined the figure and legend more clearly. C= control sample, BL21 (DE3) transformed with empty pET28a(+) vector
13. Page 9, 1st line: Reference to figure S21 is incorrect. S21 is the SDS-PAGE of FPLC fractions. It should only be S22.	Corrected
14. Specify BL21(DE3) in the following sentence: 'enzyme derived from a 4 L induced E. coli BL21(DE3) pET28a (+) culture'.	We have clarified this sentence in the main manuscript.
A schematic of the closed-loop stripping technique in the supporting information section would be useful for a broad audience.	Thank you, we have included a clarifying figure into the SI
15. Figure S25: Peaks are hard to distinguish. It looks like peaks corresponding to compounds 1 and 2 are seen in the negative control as well.	We appreciate the careful analysis of the GC-MS data and agree that in this figure (now Figure S26), the control appears to contain very minor additional signals with similar retention time. We have reanalyzed the data and the m/z values related to these signals do not match to those of product 22 and 23. Thus, we consider these as minor impurities and non-essential for the conclusion of this experiment (other metal ions had no significant influence on the production of drimenol 1 , compound 22 or 23).
Reviewer #3 (Remarks to the Author):	
Throughout the manuscript, it was clear and understood that the experimental methods and results strongly support	We acknowledge the time of the reviewer and are thankful for the constructive comment.

the conclusions. If the authors revise the manuscript of their paper in consideration of the following points, I will consider it suitable for publication.	
1. I think the title is somewhat inappropriate. I think that this title does not do a good job of understanding what is meant by "characterisation" and "antimicrobial properties". However, it is not a very serious issue. Please consider it.	We appreciate the comment and have adapted the title slightly.
2. In Introduction. The importance of VOCs is somewhat explained, but we thought that if the authors could actually mention examples of past "a key communication role", the point of focusing on VOCs in this study would be better conveyed.	We have revised the introduction as stated also by reviewer 1 and 2.
3. In P3L64-74. Only spectral data are enumerated in SI, and the entire actual structure determination process is omitted. It seems to me that the structure determination process should be explained, if only a little.	A paragraph explaining a structure elucidation process of compounds 2-6 has been included into the manuscript.
4. In P5 Figure 3. If hydrogen is written in the structural formula, an arrow from hydrogen to HMBC correlation should be shown.	Corrected. Main HMBC correlations of hydrogen in C-5 position were included into Figure 3.
5. In P7 Scheme 2. I don't understand why there are two pathways coming up with FPP and epoxy-FPP as substrates. For each step of the conversion, they can only rely on the cited literature, and it is not clear how this is supported by past findings.	We apologize for not clarifying the context. It has been described in the literature that type II terpene cyclases cyclise epoxidized substrates (e.g. oxidosqualene cyclase) yielding the respective cyclized skeleton and the hydroxy-group at the respective position. As isolated compound 4 of 5 carry a hydroxy group at the respective position, we speculate that this might either derive from a dedicated oxidizing enzyme after cyclization of FPP, or from cyclizing epoxy-FPP.
6. In P7 the explanation of Figure 4 (To test ... were detectable.). The description of the methods and results of bioinformatics analysis is too brief to understand the content. Please describe in more detail and carefully.	The sentence phrasing was misleading. In a previous study we identified Termitomyces terpene cyclases from different strains, that could be categorized according to their putative cyclization pathway. None of these groups fit to drimenol or isolongifolene biosynthesis, so none of the previously

	identified enzymes could be candidates for biosynthesis of drimenol or isolongifolene. In a next step we performed BLAST search of a characterized ascomycete drimenol synthase against predicted proteins from selected Termitomyces strains.
7. In P8. The description of the RNA-seq results is too brief to understand.	The analysis was already reported in a previous study (10.1128/msystems.01214-21) with full details in the analytical section. We have included the reference with the manuscript as well as within the Supporting Information The reference
NMR spectra:	Copies of NMR spectra have been provided for all new and known compounds in a Supplementary Data file, separate from the main Supplementary Information PDF.

REVIEWERS' COMMENTS:

Reviewer #2 (Remarks to the Author):

The authors have addressed the critical points raised by the reviewers and have made significant changes to the manuscript. The additions of useful background information along with detailed figures and mechanisms in the paper and the supporting information, and a clear discussion of their results and conclusion have vastly improved the manuscript. The authors have made a convincing case for their results being suitable for a Communications Chemistry publication. The manuscript is therefore recommended for publication with a few minor revisions.

1. Abstract first sentence: Macrotermite termites have farmed fungi in the genus *Termitomyces* as a food source since millions of years.

2. Page 10 in the revised manuscript (Word document), ...heterologous production of the histidine-fusion (His6) proteins was performed in *E. coli* BL21 using a pET28a vector : BL21 and BL21(DE3) are two different strains. Specify that it's a BL21(DE3) strain.

Reviewer #3 (Remarks to the Author):

The authors responded appropriately to my points. In particular, the revision of the INTRODUCTION section will make it easier for readers to understand the significance of this paper. I look forward to seeing this paper in the published version.